# The Perception and Intervention of Internship Nursing Students Helping Smokers to Quit: A Cross-Sectional Study in Chongqing, China

**DOI:** 10.3390/ijerph16203882

**Published:** 2019-10-13

**Authors:** Li Zhang, Yanhan Chen, Yalan Lv, Xia Yang, Qianyu Yin, Li Bai, Yaling Luo, Manoj Sharma, Yong Zhao

**Affiliations:** 1College of Medical Informatics, Chongqing Medical University, Chongqing 400016, China; zhangli@hospital.cqmu.edu.cn (L.Z.); yalanlv@126.com (Y.L.); 2017111298@stu.cqmu.edu.cn (L.B.); 2College of Nursing, Chongqing Medical University, Chongqing 400016, China; 0501nancy@163.com (Y.C.); 15856073966@163.com (X.Y.); 2017222514@stu.cqmu.edu.cn (Q.Y.); 3Department of Behavioral & Environmental Health, School of Public Health, Jackson State University, Jackson, MS 39213, USA; manoj.sharma@jsums.edu; 4School of Public Health and Management, Chongqing Medical University, Chongqing 400016, China

**Keywords:** tobacco dependence treatment, smoking cessation, interventions, nursing intern student

## Abstract

*Background*: Smoking is among the most preventable causes of death globally. Tobacco cessation can lessen the number of potential deaths. The China Tobacco Cessation Guidelines encourage medical staff to perform the 5As (Ask, Advise, Assess, Assist, Arrange) when delivering tobacco dependence treatments to patients. Nursing students will develop to be nurses in the future and they have to finish 9 months of clinical practicum study in the last year at hospitals or care centers. However, the frequency of behaviors used to help smokers quit among Chinese nursing internship students is unclear. This study analyzed the rate of nurse interns’ performance of the 5As and which demographic characteristics, perceptions of smoking and knowledge predicted higher performance of the 5As. *Methods*: The cluster sampling method was used to select 13 teaching hospitals among 29. All nursing intern students were expected to finish the questionnaire about their 5As behaviors to help patients quit smoking. Their 5As performances were scored from one to five with 5 being the best and scores were summed. A multivariate linear mixed-effect model was employed to test the differences between their 5As. *Results*: Participating in the survey were 1358 interns (62.4% response rate). The average scores were as follows—Ask—3.15, Advise—2.75, Assess—2.67, Assist—2.58 and Arrange—2.42. A total of 56.3% students perceived that medical staff should perform the 5As routinely to help patients quit smoking. On the other hand, 52.1% viewed clinical preceptors as role models of the 5As. School education regarding tobacco control, smoking dependence treatment, self-efficacy and positive intentions were predictors of higher performance of the 5As (*p* < 0.001). *Conclusions*: Nursing internship students seldom administered tobacco dependence treatments to patients. It is essential to improve the corresponding education, skills and self-efficacy of the 5As. Meanwhile, clinical preceptors should procure more training in the responsibilities and skills related to tobacco cessation. In this way, clinical preceptors can be role models of the 5As and impart positive influences on interns.

## 1. Introduction

Tobacco use is among the most preventable causes of death. It kills nearly 8 million persons each year, worldwide [1]. Smoking also significantly increases the social–economic burden in the form of health care costs and lost productivity [2]. 

China is the largest producer and consumer of tobacco in the world, accounting for 40% of global cigarette production and consumption [3]. More than one million Chinese die each year from tobacco-related diseases [4]. Tobacco cessation is an effective way to eliminate tobacco hazards. China has 316 million current smokers. The smoking rate for people over 15 years old is 27.7%–52.1% in males and 2.7% in females. Only 17.6% report an intention to quit smoking and 7% plan to quit within one month [5].

Studies have demonstrated that evidence-based tobacco dependence treatment is one of the effective ways to reduce tobacco harm. Advice from physicians [6,7,8] and nurses [9] is often helpful in inducing the willingness to receive tobacco dependence treatment and improve the success rate. They also found nonsmoking healthcare professionals to be more positive in tobacco cessation help [10]. The China Clinical Tobacco Cessation Guidelines claim that nurses deliver the 5As to patients to quit smoking [11]. The 5As are—Asking patients about smoking history, Advising smokers to quit smoking, Assessing smokers’ willingness to quit smoking, Assisting with developing a quit plan and Arranging follow-up contacts related to smoking. China has 4 million registered nurses [12] and the largest team of health professionals, who are all predominantly nonsmokers. Therefore, non-smokers comprise the majority of healthcare professionals and play pivotal roles in curbing the tobacco epidemic [10,13,14,15,16]. Nonetheless, previous studies have indicated that few clinical nurses have assisted patients in quitting [13,14]. Sarna et al. found that 64% nurses in Beijing and Hefei asked about smoking history, 52% assessed smokers’ willingness to quit and 17% arranged follow-up contacts related to smoking [14].

Limited training was the main hindrance to delivering tobacco dependence treatments for nurses [13,14]. Nurses who were trained in tobacco treatment were more likely to use and follow tobacco cessation guidelines than those who were untrained [17,18,19]. Medical school education is the optimal time for this training [20]. Fewer than 40% of respondents reported having received formal tobacco cessation education in a global health student survey [21]. Tobacco use also influenced nursing students’ attitudes and behaviors toward smoking control [22].

China has a majority of female nursing students (95%), whose rate of smoking (2.9%) is lower than that of males [23]. Little attention has been paid to the 5As treatment behaviors of nursing students and their tobacco cessation education. However, tobacco sales are newly targeting women and youth (with the former receiving more attention and typically aged 15–24 [24]). A total of 8.5% nursing students reported a distinct tendency to smoke when invited by good friends and 13.2% would smoke during social situations in the future [25].

Tobacco cessation education among nursing students is a significant means to reduce their tobacco use, improve their self-efficacy and enhance their tobacco cessation capabilities [26,27]. This education is not only very important for their own health but may also inspire them to have an extensive post-graduate impact on the health education and health promotion among the public. Nevertheless, studies have suggested that over 25% Chinese nursing schools do not have any knowledge about tobacco cessation and 31% schools delivered no more than 1 h of tobacco cessation instruction, only about half of which mentioned the 5As [13,28].

Healthy China 2030 Planning Outline set a smoking control target to lower the rate of smokers aged over 15 years from the 27.7% (at present) to 20% by 2030 [29]. An effective way to contribute to achieving this goal is to include tobacco cessation courses in medical schools and train tobacco cessation professionals. However, the corresponding course education among nursing students remains unreported. Over 200,000 Chinese nursing students graduate each year [30] and they are a very important component of delivering tobacco cessation treatment. Chongqing, the municipality with the largest population and more rural citizens, features a high smoking rate and extensive second-hand smoking exposure [5]. Therefore, more medical staff are needed to help people quit smoking and lessen the associated tobacco hazards.

Chinese nursing students in their fourth year need to take a clinical practicum no less than 9 months (40 weeks) before graduation, which is a good time to determine a students’ tobacco cessation education and evaluate its effects. This study, conducted in Chongqing, the largest city in western China, investigated the knowledge, attitudes and education regarding tobacco cessation and 5A behaviors that help one to quit smoking among nursing internship students. The 5As differences were analyzed on the basis of demographics, education and the perception of tobacco cessation involvement.

## 2. Materials and Methods 

### 2.1. Study Design and Participants

A cross-sectional survey of the extent to which nursing internship students help smokers quit was conducted in some Teaching Hospitals in Chongqing. The sample size was calculated using the formula N = K * [Z^2^_a/2_ * π (1−π)/δ]. Sarna et al. [14] found that 52% of nursing interns assessed smokers’ willingness to quit and offered help. The error δ was 3% and the confidence degree Z_a/2_ was 95% with 1066 participants. The study employed a convenient online survey because nursing interns were scattered among hospitals and units. Because of the usual low retrieval rate of online questionnaires, the survey used the cluster random sampling method with a random number table. It investigated the targeted students in 13 teaching hospitals from 29 that were in compliance with inclusion criteria. Inclusion criteria of the hospitals were—(1) general hospital, (2) teaching hospital, (3) more than 400 beds, (4) training nursing internship students, (5) being in Chongqing. Inclusion criteria of the nursing interns were—(1) being a nursing student, (2) engaging in a clinical internship following on-campus theoretical study, (3) being a non-registered nurse and (4) being a voluntary participant. 

### 2.2. Questionnaire

The questionnaire was designed based on the knowledge–attitude–practice (KAP) Model. The first iteration was drafted from the literature and tailored to the interviewees. The final version was identified after a few discussions with the expert panel, expert members from tobacco dependence treatment, public health, statistics, nursing education and clinical nursing from abroad and home. The Cronbach’s alpha (demographic characteristics were excluded) was 0.952 including 32 items in 4 parts.

#### 2.2.1. Demographics

Demographic characteristics included gender, age, education, clinical internship duration and whether one was a current smoker. Current smokers are those who smoked for at least 1 day in the last 30 days [31,32].

#### 2.2.2. Knowledge and Attitude about being Involved in Tobacco Cessation

(1) Tobacco-associated knowledge—Students answered 34 questions, from smoking prevalence (6 questions) to advice on clinical tobacco treatment guidelines, including 5 questions about second-smoking harm on families, 10 about knowledge of tobacco and disease, 9 about banning smoking in public areas and 4 about resources for tobacco cessation. One correct answer was calculated for 1 score and the gross score was 34. (2) Attitude: 1) Whether medical staff should routinely perform the 5As to help smokers quit; 2) whether the preceptor/mentor was a role model in helping smokers quit. Response choices were calculated as: 1 = strongly disagree, 2 = disagree, 3 = neutral, 4 = agree and 5 = strongly agree. (3) Self-efficacy in tobacco counseling. Students answered the question “How proficient are you in each of the 5As (Ask, Advise, Assess, Assist and Arrange)?” Response choices were calculated as: 1 = unskilled, 2 = somewhat skilled, 3 =skilled and 4 = very skilled. The summed scores ranged from 5 to 20. 

#### 2.2.3. Tobacco Use Education

(1) Students reported their classroom time spent on tobacco-related education at nursing school as 0 minutes, 30 minutes, 30–60 minutes, 1–2 h, 2–3 h, 3–4 h, 4–5 h or >5 h. Teaching methods were a. instruction, b. case discussion, c. scenario simulation, d. skills training and e. web-based education. The response choices were “yes” or “never”.

(2) Students reported the application of the 5As in their internship as: 1) the number of times the 5As were performed by the preceptor and 2) the number of times the participants were instructed to perform the 5As. Response choices were calculated as: 1 = none, 2 = 1 to 3 times, 3 = 4 to 9 times, 4 = 10 to 25 times and 5 = >25 times. The summed scores ranged from 5 to 25.

#### 2.2.4. Behavior

Students shared the number of patients whom they had performed the 5As on to help them quit smoking during their nursing internship—a) Asking about smoking history, b) Advising to quit smoking, c) Assessing willingness to quit, d) Assisting with developing a quitting plan and e) Arranging follow-up contacts related to smoking. Response choices were calculated as: 1 = none, 2 = 1 to 3 patients, 3 = 4 to 9 patients, 4 = 10 to 25 patients and 5 = >25 patients. The summed score ranged from 5 to 25.

### 2.3. Survey Methods

The data were collected anonymously through an online investigation. The investigation was approved by the dean of the nursing department and was discussed with the internship manager regarding the objectives and methods of the survey. Nursing interns completed the questionnaire with a mobile phone or computer after being made clear of the survey’s objectives and methods by way of QQ or WeChat.

### 2.4. Quality Control

The questionnaire was revised from the literature [31,32,33] and modified several times after discussions with experts and two preliminary investigations. Eligible questionnaire would be rewarded with ¥1 attached to the questionnaire. Instructions and phone numbers for the researchers were available at the beginning of the questionnaire in case of any doubts. Furthermore, no repeated submissions were allowed from one Internet Protocol Address. Before being imported into the database, the data were carefully processed with the Excel software. Invalid questionnaires were deleted (those completed by participants who did not meet the inclusion criteria and those finished in less than 180 seconds.

### 2.5. Ethical Approval

The Institutional Review Board at the First Affiliated Hospital of Chongqing Medical University approved the study before the implementation was recorded (2019–157) and this approval was extended to the hospitals. An online consent form explained the aim of the research.

### 2.6. Statistical Analysis

Data from 1358 nursing internship students were analyzed with SPSS20.0 (IBM Corporation, Armonk, NY, USA). Characteristics of the participants were expressed and descriptively described with means and standard deviations or frequencies and percentages. The frequencies of the 5As behaviors were calculated in scores. Because they did not satisfy the parameter hypothesis test condition, the 5As behavior-related factors were analyzed with the Wilcoxon rank sum Test and the Kruskal–Wallis Test. The study dichotomized the knowledge score and self-efficacy score according to the median, also dichotomized the instruction frequency and observation frequency according to the median. Considering the possible correlation among the participants in the same hospital, the factors attributable to the internal correlation of complex samples were analyzed by cluster unit analysis of hospitals. The factors affecting the implementation of 5A behavior of students were analyzed by multi-factor linear mixed effect model. The summed scores of 5As behavior implementation were the dependent variable and the independent variables covered students’ primary information, smoking cessation-related perceptions and education. The significance threshold of these statistics was 0.05.

## 3. Results

### 3.1. Demographic Characteristics and the 5As Implementation

In total, 2440 nursing internship students from 13 teaching hospitals were eligible for the survey and 1522 students completed the questionnaire. The overall response rate was 62.4%, generated from a range of 46.6% to 78.8%. As mentioned earlier, 1358 questionnaires were put to use after the removal of 146 responses with a response time less than 180 seconds and three from graduate students and 15 from areas beyond Chongqing.

Demographic characteristics—Table 1 describes the 5As behaviors associated with characteristics. Students were aged 16–32, with a mean of 20.15 ± 1.920; 101 were male (7.4%) and 1257 were female (92.6%). A total of 36 of the respondents were current smokers (2.7%); 19 were male (18.8%) and 17 were female (1.4%). About half (47.1%) had associate degrees. Students with secondary school education were more likely to deliver 5A interventions than those with associate degrees, while those with bachelor’s degrees offered the least 5As. The 5As frequency differences arising from these education differences was statistically significant (*p* < 0.001). The internship students studied in hospitals for more than 6 months. Along with this time, the longer the internship was, the more frequently the 5As were implemented (*p* < 0.001). Regarding 5As implementation, the tobacco dependence treatment gross score ranged from 5 to 25 (M = 13.58, SD = 5.556). The average scores, respectively, were—Ask—3.15, Advise—2.75, Assess—2.67, Assist—2.58 and Arrange—2.42. The 5As scores of those who reported offering tobacco dependence treatments to more than 10 patients were Ask—38.4%, Advise—26.3%, Assess—24.4%, Assist—22.2% and Arrange—19.4%. 

### 3.2. Students’ Knowledge, Attitudes and 5As Implementation

Table 2 describes tobacco cessation knowledge and related attitudes and 5As implementation. The summed score of knowledge was 34 and students scored 5–33 (23.14 ± 4.47). The higher the knowledge score, the higher the frequency of the delivery of the 5As (*p* < 0.001). About 74.6% students disagreed that low-tar and low-nicotine cigarettes were less harmful to one’s health. Further, 37.8% thought smoking was a chronic disease and should be treated with drugs. Knowledge scores on the harm of secondhand smoke to families and children were—asthma attack or asthma exacerbation of families and children (91.7%), fetal death or malformation (92.3%), pneumonia in children (88.4%), miscarriage or preterm birth (85.3%) and coronary heart disease (71.5%). The majority believed that smoking led to lung cancer (97.9%), tooth damage (97.3%), premature birth or fetal malformation (95.8%), emphysema (82.9%) and peptic ulcers or tumors (80.1%). More than half knew that smoking triggered coronary heart disease (78.1%), stroke (70.6%), vasculitis (76.8%), male dysfunction (76.1%) and osteoporosis (67.4%). Most thought that schools (96.5%) and hospitals (98.5%) should ban smoking, while few thought that internet cafes (31.1%) and bars (24.8%) should also ban smoking. Tobacco cessation resources—8.9% were familiar with China Clinical tobacco cessation Guideline. 10.3% knew tobacco cessation drugs and 8.3% knew tobacco cessation website, while 6.8% spoke out cessation hotline immediately. 

Attitudes towards involvement in tobacco cessation—About 56.3% thought that medical staff should use the 5As routinely to administer tobacco dependence treatments. 52.1% agreed that their preceptors are role models in helping smokers quitting smoking. 

Confidence of 5As Implementation—The percentage scores of students who self-reported being skillful or very skillful in the 5As were as follows—Ask—53.5%, Advise—38.7%, Assess—41.7%, Assist—34.1% and Arrange—33.4%.

The Relationship between the Perception of Tobacco Cessation and 5A Behaviors—The higher the tobacco-related knowledge and the more positive their attitudes towards the implementation of the 5As and the higher their self-efficacy, the more frequently implemented the 5As (*p* < 0.001), as seen in Table 2.

### 3.3. Tobacco Cessation Education and 5A Implementation 

Table 3 displays the tobacco cessation education that students obtained. During school study before the internship, 30.6% students reported that they never received tobacco cessation education. Most participants (46.6%) received education for less than 1 hour. Learning forms were classroom instruction (45.9%), case discussion (39.7%), web-based learning (37.1%), scenario simulation (29.6%) and skill training (23.2%). The 5As scores of tobacco treatment observation were 5–25 (M = 12.71, SD = 5.749). The overall scores were relatively low. Students who observed the 5As treatment from the preceptors over 10 times scored as follows—Ask—24.7%, Advise—22.7%, Assess—22.5%, Assist—20.7% and Arrange—19.3%. Those who never observed the 5As from their teachers amounted to 18.0%–27.8%. The average score for receiving 5A instruction from preceptors was 13.08 (SD = 5.683). The scores for students who received 5A instruction more than 10 times were—Ask—30.9%, Advise—25.2%, Assess—23.4%, Assist—22.1% and Arrange—20.0%. Those who had never been instructed about the 5As by preceptors amounted to 13.7%–27.6%. The longer the school tobacco cessation education, the more common the observations of 5A behaviors from their preceptors, the more frequent the 5A instruction from their teachers and the more 5A behaviors students delivered to their patients (*p* < 0.001, with positive Spearman correlation).

### 3.4. Impact Factors of the Frequency of Tobacco Dependence Treatment Behaviors

Considering the possible associations among the participants in the same hospital, a mixed-effect model was employed to examine the multivariate factors affecting implementation of the 5As among nursing internship students. Significant factors affecting the frequency of 5As implementation included more tobacco treatment knowledge (95%confidence interval (CI): 0.1432, 0.9389, *p* < 0.001), higher tobacco treatment self-efficacy (95%CI: 0.7376,1.6208, *p* < 0.001), positive attitude towards the routine use of 5As skills (95%CI: 0.2336, 1.1452, *p* < 0.001, perception of teachers as the 5A behavior models (95%CI: 0.6436,1.6180, *p* < 0.001), longer school tobacco cessation education (95 CI: 2.2942, 3.3046, *p* < 0.001) and more 5As per performance observed from teachers (95%CI: 3.0646, 4.0824, *p* < 0.001 (seen in Table 4).

## 4. Discussion

This study found that the 5A behaviors of nursing internship students helping smokers quit were associated with tobacco cessation education [18] and the preceptors’ 5A behaviors, as well as tobacco knowledge and 5As self-efficacy. This result is consistent with that of previous studies [33,34,35,36], as well as Behavior Change Theory such as the social cognitive theory and the theory of planned behavior [37,38,39,40,41,42]. 

This study further revealed that nursing internship students in Chongqing scarcely employed the 5As to help smokers quit. At the time of the survey, 95% participants had studied more than 20 weeks in clinical and cared for over 100 patients with the help of preceptors. However, their Ask score, the most frequent 5A behavior, was only 3.15. Thus, they only asked around 10 patients about their smoking history, while the other items of the 5As were performed on less than 10 patients. The lack of tobacco cessation education and unawareness of the related resources possibly contributed to the lower utilization of tobacco cessation resources. The survey found that one in three nursing internship students never received tobacco cessation education and half obtained less than 1 h of education, which is similar to outcomes 10 years ago [13]. Interns accumulated more knowledge about tobacco cessation than on-campus students [43] but they still knew little about the smoking quit hotline, smoking cessation drugs and tobacco cessation guidelines. Studies report that the enhancement of behaviors that help others quit smoking requires knowledge and skills of tobacco cessation, as well as the recognition and effective utilization of related resources [44]. This suggests that future education needs to instruct students about how to access tobacco cessation resources and how to be well-prepared with the corresponding knowledge, awareness and competencies. 

Low tobacco cessation awareness can be ascribed to a lack of tobacco education. This study revealed that no more than 60% of nursing students thought that medical staff should routinely implement the 5As to help patients quit smoking; this result was worse than that for the medical students reported in the literature [32]. Medical students received more attention in China, possibly because the majority of medical students were males who had a higher smoking proportion [45]. Nursing students have been neglected, perhaps due to their female gender and lower smoking rate [23]. It is, therefore, unsurprising that they thought quitting smoking was a sort of tobacco cessation effort and were not aware that helping others to quit smoking was the responsibility of all healthcare professionals. Therefore, promoting 5As performance among nursing students might be improved through increasing tobacco cessation education, thereby enhancing their cognition of the harm of tobacco and the self-efficacy of their tobacco cessation involvement, as well as 5As skills and good utilization of related resources. Then, nursing students would employ more 5A behaviors. 

This study also found the 5As treatment behaviors were similar to those of their teachers. If they thought their teachers were models of the 5As, they had stronger intentions to deliver appropriate treatment services (similar to the results from the literature [33]). Clinical preceptors should improve their smoking cessation knowledge and treatment behavior training to become role models. In this way, they can help students to raise awareness and develop 5As treatment behaviors. However, this study revealed that preceptors in clinical environments rarely performed the 5As on their patients and imparted little instruction of the 5As to their students.

This was the first survey to investigate 5As implementation among nursing interns in Chongqing, China. We were fortunate to be able to collect data from more than 1300 students in 13 teaching hospitals. We had tried to clarify the situation that nursing internship students helped patients quit smoking and their corresponding education at school. This study indicated that it was urgent to implement tobacco cessation education, improve the responsibility of tobacco cessation and develop 5As skill training among nursing students. Teaching hospitals should offer more training to enhance clinical preceptors’ 5As skills and behaviors. 

This study has several limitations. Firstly, because the working hours of internship students were scattered, we could only take a network survey, which was convenient for the students. It is known that the response rate of the network survey was low (62.4%) and we were unsure of the clinical implementation of the 5As for students who did not submit. To ensure the effectiveness of the study, we removed questionnaires completed within 180 seconds. To some extent, our data represent the current situation for nursing internship students in Chongqing in helping smokers. Secondly, our survey was conducted in Chongqing, an undeveloped western city in China. Although we benefited from a relatively large sample size, the implementation of tobacco cessation might not be as popular as that in developed cities and regions [46]. A future survey could study the national status, with a larger sample size used to cover cities with different economic and cultural levels. Thirdly, the data were collected in hospitals and we were uncertain about whether the differences of the 5As behaviors resulted from on-campus school education or not because hospitals enrolled students from different schools with different education backgrounds and questionnaires were completed anonymously. We could only differentiate between different classroom education durations. Further studies can take the form of a self-report for students about their schools to identify the impact of school on tobacco cessation behaviors. Fourthly, the findings from the self-reporting may have been overestimated because of social expectations, especially among the 5A behaviors in clinical. Furthermore, this study may have neglected the impact of patients’ smoking on the 5A behaviors of students because it focused on the investigation of the relationship between the 5A behaviors in tobacco dependence treatment and personal characteristics of students. Lastly, the data were obtained from a across-sectional survey and need to be put into practice to be tested.

## 5. Conclusions

Nursing student interns rarely implement the 5As to help patients quit smoking and know little about accessible tobacco cessation resources. There should be more on-campus tobacco cessation education on the knowledge as well as implementation of the 5As to improve the students’ self-efficacy. Clinical preceptors also should be provided with more education on tobacco cessation and corresponding skills training, as they need to be models of the 5As to be a positive influence on students.

## Figures and Tables

**Table 1 ijerph-16-03882-t001:** Relationship between primary characteristics and implementation of the 5As.

Demographics	All the Students	The Score of Using 5A		
	*n* (%)	Median (P_25_, P_75_)	H/Z	*p*
Age: (mid-value 20)			25.637	0.000 **
≤17	158 (12.60)	15.00 (10.00, 19.00)		
18	187 (14.80)	15.00 (10.00, 19.00)		
19	91 (7.10)	14.00 (10.00, 17.00)		
20	254 (18.70)	12.00 (10.00, 18.00)		
21	324 (23.00)	12.00 (10.00, 16.00)		
22	247 (17.30)	12.00 (10.00, 16.00)		
≥23	97 (6.50)	11.00 (9.00, 15.00)		
Gender: % (N)			−1.931	0.053
Male	101 (7.40)	15.00 (10.00, 18.00)		
Female	1257 (92.60)	13.00 (10.00, 17.00)		
Professional education			33.88	0.000 **
Diploma	325 (23.90)	15.00 (10.00, 20.00)		
Assoc. degree	639 (47.10)	13.00 (10.00, 17.00)		
Baccalaureate	394 (29.00)	11.00 (10.00, 15.00)		
Current smoker: N (%)			−0.455	0.649
Yes	36 (2.70)	13.00 (10.00, 15.00)		
No	1322 (97.30)	13.00 (10.00, 17.00)		
Internship time			19.06	0.001 *
4 months	61 (4.50)	10.00 (9.00, 15.00)		
5 months	431 (31.70)	13.00 (10.00, 17.00)		
6 months	600 (44.20)	13.00 (10.00, 16.50)		
7 months	209 (15.40)	15.00 (10.00, 19.00)		
≥8 months	57 (4.20)	15.00 (10.00, 18.00)		

Note: Wilcoxon rank sum Test was used, * *p* < 0.05, ** *p* < 0.001 (statistically significant).

**Table 2 ijerph-16-03882-t002:** Relationship between students’ attitudes and the implementation of the 5As.

Variables	All the Students	The Score of Using 5A		
	*n* (%)	Median (P_25_, P_75_)	Z	*p*
Knowledge score (median 24)			−4.280	0.000 **
<24	631 (46.5)	12.00 (10.00, 16.00)		
≥24	727 (53.5)	14.00 (10.00, 18.00)		
Medical staff routinely use 5As to treat smokers			−11.121	0.000 **
Disagree	593 (43.7)	10.00 (9.00, 15.00)		
Agree	765 (56.3)	15.00 (10.00, 20.00)		
Tobacco self-efficiency of 5As (5–20 score, median 11)			−16.002	0.000 **
<11	616 (45.4)	10.00 (8.00, 1400)		
≥11	742 (54.6)	15.00 (11.00, 20.00)		
Teachers are models in the 5As implementation			−15.721	0.000 **
Disagree	650 (47.9)	10.00 (8.00, 15.00)		
Agree	708 (52.1)	15.00 (10.50, 20.00)		

Note: Wilcoxon rank sum Test was used, ** *p* < 0.001 (statistically significant).

**Table 3 ijerph-16-03882-t003:** Relationship between tobacco cessation education and the implementation of the 5As.

Variables	All the Students	The Score of Using 5A		
	*n* (%)	Median (P_25_, P_75_)	H/Z	*p*
Classroom instruction time			284.482	0.000 **
0	415 (30.60)	10.00 (7.00, 15.00)		
30 min	362 (26.70)	11.00 (10.00, 15.00)		
30–60 min	270 (19.90)	14.00 (10.00, 17.00)		
1–2h	147 (10.80)	15.00 (11.00, 18.00)		
2–3h	57 (4.20)	17.00 (13.00, 20.00)		
3–4h	41 (3.00)	20.00 (15.00, 20.00)		
>4h	66 (4.90)	25.00 (20.00, 25.00)		
Total 5As instruction frequency (5–25 score, median 12)			−24.481	0.000 ^**^
<12	640 (47.10)	10.00 (7.00, 11.00)		
≥12	718 (52.90)	16.00 (14.00, 20.00)		
Total 5As observation frequency (5–25 score, median 11)			−23.471	0.000 **
<11	663 (48.80)	10.00 (7.00, 11.00)		
≥11	695 (51.20)	16.00 (14.00, 20.00)		

Note: Wilcoxon signed rank sum test was used, ** *p* < 0.001(statistically significant).

**Table 4 ijerph-16-03882-t004:** Multivariate linear mixed effect model was used to predict tobacco treatment 5A behavior of nursing students.

Parameters	Estimation	Standard Error	*p*	95% CI	
				Lower Limit	Upper Limit
Gender					
Female	reference				
Male	−0.206	0.398	0.606	−0.987	0.576
Age					
≤17	reference				
18	0.336	0.400	0.401	−0.448	1.119
19	0.856	0.532	0.108	−0.188	1.899
20	−0.038	0.469	0.935	−0.959	0.882
21	−0.047	0.487	0.923	−1.002	0.908
22	−0.192	0.524	0.714	−1.220	0.836
≥23	0.004	0.599	0.995	−1.171	1.179
Professional education					
Diploma	reference				
Assoc. degree	−0.444	0.406	0.274	−1.241	0.353
Baccalaureate	−0.541	0.497	0.277	−1.518	0.437
Current smoker					
No	reference				
Yes	0.406	0.651	0.533	−0.871	1.683
Internship time					
4 months	reference				
5 months	−0.200	0.520	0.701	−1.220	0.820
6 months	0.133	0.530	0.802	−0.908	1.173
7 months	0.210	0.582	0.719	−0.934	1.353
8 months	0.565	0.684	0.409	−0.777	1.907
Tobacco knowledge					
<24	reference				
≥24	0.537	0.206	0.009*	0.133	0.940
Tobacco self-efficiency of 5A					
<11	reference				
≥11	1.213	0.228	0.000 **	0.765	1.660
Medical staff routinely use 5As to treat smokers					
Disagree	reference				
Agree	1.153	0.252	0.000 **	0.659	1.647
Teachers are models of 5As implementation					
Disagree	reference				
Agree	0.694	0.236	0.003*	0.231	1.156
Classroom instruction time					
0	Reference				
30min	0.617	0.269	0.022 *	0.089	1.145
30–60min	0.868	0.298	0.004 *	0.282	1.453
1–2h	0.855	0.368	0.020 *	0.133	1.576
2–3h	1.488	0.534	0.005 *	0.441	2.535
3–4h	2.301	0.619	0.000 **	1.087	3.515
>4h	5.633	0.516	0.000 **	4.620	6.645
Total 5A instruction frequency					
<12	reference				
≥12	3.558	0.263	0.000 **	3.041	4.074
Total 5A observation frequency					
<11	reference				
≥11	2.793	0.261	0.000 **	2.281	3.305

Note: * *p* < 0.05, ** *p* < 0.001 (statistically significant).

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
