# Peer review of "The Perception and Intervention of Internship Nursing Students Helping Smokers to Quit: A Cross-Sectional Study in Chongqing, China"

_ijerph, 2019, doi:10.3390/ijerph16203882_

Round 1
Reviewer 1 Report
Please see attached

Author Response
We appreciate your warm work. We have resised the manuscript based on your advice, please see the attachment.

Reviewer 2 Report
First of all, I would like to comment that I found the whole article very interesting, using the 5 As model.
However, I would like to know more about the 5 As model. Has it been used in any other studies, especially in China?
In addition to this, the questionnaire (lines 115 -120) should be described in detail. How many questions, sessions devided, etc. What exactly measures (nursing students knowledge)?
Also, table 3 should be properly constructed.
Author Response
We appreciate your warm work. We have revised the manuscript based on your advice, please see the attachment.

Reviewer 3 Report
See attached.

Round 2
Reviewer 1 Report
The study has improved significantly. I have a few minor comments:
Abstract
This line – “This study analyzed the 5 As associated with demographic characteristics, the
perception of smoking, and the knowledge of tobacco among those interns.” Should be changed to:
“This study analysed the rate of nurses intern’s performance of the 5As and which demographic characteristics, perceptions of smoking and knowledge predicted higher performance of the 5As.
This line “Their 5 As performances were scored from one to five, with 5 being in best” should be moved to the methods in the abstract. It should also be mentioned that a composite score of all the 5As was calculated.
Add response rate to the abstract: “1358 interns participated in the survey (62.4% response rate)”
For this line: “School education regarding tobacco control, smoking dependence treatment, self-efficacy, and positive intentions were predictors” add that they were predictors of higher performance of the 5As.
The last line in the abstract add “role” right before “models” - ”clinical preceptors can be role models…
Throughout the abstract and manuscript it should be the 5A’s without any space between the 5 and A.
Introduction
This sentence “People over 15 encounter a smoking rate of 27.7%—52.1% male and
2.7% female. Only 17.6% report the intention to quit smoking, and 7% plan to start within one month” should be edited as follows: The smoking rate for people over 15 years old is A 27.7%—52.1% in males and
2.7% in females. Only 17.6% report an intention to quit smoking, and 7% plan to quit within one month”
Line 55 – History should be history (not with a capital H).
Line 58 – “who are all predominantly nonsmokers”
Line 84 - should be edited as suggested: “An effective way to contribute to achieving this goal is to include tobacco cessation courses in medical schools and train tobacco cessation professionals.”
Methods
Line 106 – should be the low retrieval rate….
Line 110 – no need to include the exclusion criteria if they are simply the reverse from the inclusion criteria. You can delete the exclusion criteria for both the hospital and students.
Line 125 – add that the final survey (translated to English) can be found in the supplemental file. In my opinion no need to include the first versions and the expert panel discussions – just provide the final version.
Line 156 – “the general score” should be changed to “the summed score” for consistency.
Line 172 – you mean 180 seconds (not minutes)
Line 186 – “The gross scores of 5 As behaviour implementation were the dependent variable…” – was it not the summed score that was the dependent variable? You did not perform the model for each component of the 5As by itself, right?
Results
The first paragraph – the numbers do not add up. 2440 were eligible, 1522 answered, 149 were excluded = 1373 and not 1358 – what happened to 15 surveys?
Also this sentence “The percentage of the effective questionnaire was 89.2%.” is not important and can be deleted.
Line 214 – In the methods you mention that for the knowledge score you calculated a percentage from correct answers, but here you provide a result which simply provides the number of correct answers. If it’s a percentage the result should be between 0-100%, so if they scored between 5-33, then their score should be between 15%-97%. It’s OK to leave it as their score out of 34 was 5-33, but you need to correct this in the methods. Also, no need to repeat that the total score was out of 34 as this was mentioned in the methods.
Line 227, 230, 233 –place a colon (“:”) at the end of these as they are subheadings.
Line 228 – “52.1% took the preceptors as models” – Do you mean “52.1% agreed that their preceptors are role models in helping smokers quitting smoking”
In the methods you need to add that for the Wilcoxon rank tests exploring the relationship between students’ attitudes and the implementation of the 5As (table 2), you dichotomized the knowledge score, and self-efficacy score according to the median. Same for the model exploring the relationship between tobacco cessation education and the implementation of the 5As (table 3) – you dichotomized the instruction frequency and observation frequency according to the median.
Line 234 – “The greater the perception of tobacco-related knowledge, the more positive their attitudes towards the implementation of the 5As.” – do you not mean the higher their tobacco-related knowledge, the more frequently implemented the 5A’s? (the dependent variable is the implementation, not the attitudes towards the 5As or am I mistaken?)
Line 272 – “as well as Behavior Change Theory” – please add “such as the social cognitive theory and the theory of planned behaviour”.
Line 278 – “The lack of tobacco cessation education and unawareness of the related resources resulted in bad utilization of tobacco cessation resources” – a cross- sectional study can never show causality therefore findings are always suggestive of… the sentence should be revised - “The lack of tobacco cessation education and unawareness of the related resources possibly contributed to the lower utilization of tobacco cessation resources”
Line 282 – “but they still knew little about the smoking quit hotline, smoking cessation drugs, and tobacco cessation guidelines.” – you have not provided results for this in the results section – you provided the overall knowledge score and results for the specific questions on harms. If you want to mention this in the discussion, you should provide these results specifically – what proportion of participants were familiar with the medications? Quitline? And so forth.
Line 296 – this whole paragraph should be rewritten to be more suggestive. You did not prove this and you don’t know this for sure. This is a suggestion…
“Therefore, promoting 5 As performance among nursing
students might be improved through increasing tobacco cessation education, thereby enhancing their cognition of the harm of tobacco and the self-efficacy of their tobacco cessation involvement, as well as 5As skills and good utilization of related resources.
